# Hybrid Techniques to Predict Solar Radiation Using Support Vector Machine and Search Optimization Algorithms: A Review

José Manuel Álvarez-Alvarado [1], José Gabriel Ríos-Moreno [2,*], Saul Antonio Obregón-Biosca [2], Guillermo Ronquillo-Lomelí [3], Eusebio Ventura-Ramos, Jr. [2] and Mario Trejo-Perea [2]

[1] División de Investigación y Posgrado, Facultad de Ingeniería, Universidad Autónoma de Queretaro, Queretaro 76010, Mexico; jmalvarez@uaq.mx
[2] Facultad de Ingeniería, Universidad Autónoma de Queretaro, Queretaro 76010, Mexico; saul.obregon@uaq.mx (S.A.O.-B.); eventura@uaq.mx (E.V.-R.J.); mtp@uaq.mx (M.T.-P.)
[3] Centro de Ingeniería y Desarrollo Industrial, Querétaro 76125, Mexico; gronquillo@cidesi.edu.mx
* Correspondence: riosg@uaq.mx; Tel.: +52-442-192-1200 (ext. 6064)

**Abstract:** The use of intelligent algorithms for global solar prediction is an ideal tool for research focused on the use of solar energy. Forecasting solar radiation supports different applications focused on the generation and transport of energy in places where there are no meteorological stations. Different solar radiation prediction techniques have been applied in different time horizons, such as neural networks (ANN) or machine learning (ML), with the latter being the most important. The support vector machine (SVM) is a classification method of the ML that is used to predict solar radiation. To obtain a better accuracy of prediction data, search optimization algorithms (SOA) such as genetic algorithms (GA) and the particle swarm optimization algorithm (PSO) were used to optimize the prediction accuracy by searching the model parameters. This article presents a review of different hybrid SVM models with SOA applied to obtain the best parameters to reduce the prediction error of solar radiation using meteorological variables. Research articles from the last 5 years on solar radiation prediction using SVM models and hybrid SMV optimized models with SOA were studied. The results show that SVM with GA presents a better performance than the classical SVM models using the Radial basis kernel function for prediction parameters.

**Keywords:** solar radiation; support vector machine; heuristic algorithm; renewable energy; solar energy systems

## 1. Introduction

In recent years, energy generation and transport have become very important issues for the social and economic development of any nation that wants to be sustainable. Today, the demand for fossil fuel is 80% of the total energy consumed globally and more than 95% is used for the transport sector [1]. The use of these fossil fuels has been one of the main causes of the greenhouse effect on earth [2–4]. Nowadays, the scientific community has set the task of developing new technologies focused on the generation and use of electricity through solar power [5–7].

Additionally, power companies must be able to manage energy production to meet consumption at any time [8]. This is why it has focused on generating new techniques to manage energy production, as it is an important factor for a society to thrive economically and without harming the environment by using alternative energies [9]. However, alternative energies (such as solar, wind, to name a few), are difficult to represent in a mathematical model because of their non-linear behavior. In order to meet the balance between generation and consumption, it is crucial to predict solar radiation in high-capacity power generation facilities.

In this context, acquiring further knowledge on solar radiation has been one of the main research topics, for which it has become a benchmark in the strengthening of energy

generation strategies through the use of renewable energy sources. In this context, machine learning (ML) has become a recognized strategy in this field [10]. A high-performance solar energy generation system largely depends on the forecast of the output power, since this data can support the design and sizing of these systems. Under this concept, forecasting models of global solar radiation are developed under two main categories: satellite cloud images and ML models [11].

Forecasting solar radiation is becoming a popular topic. This technology allows solar energy to be integrated into the grid, producing good results by improving the quality of energy supplied to the grid to reduce the costs of accessories related to the use of this resource [12]. The combination of these factors has been the motivator for the development and design of models of a complex field of research that aims to produce better predictions of the solar resource and thus be able to predict the output power that can be generated depending on the type of technology used and the prediction horizon used [13].

A forecasting model essentially consists of a system of linear or non-linear equations that relate the future values of the variable to be predicted with recorded data variables themselves and the explanatory variables. Before making a prediction, you must define the prediction horizon on which the model should be applied [14]. According to the literature, there are two classes of techniques to choose the method according to horizon time: The Now-casting method, defined as a forecast for the next 6-hour period, based on detailed observational data such as radiometers, pyrheliometers, satellite images or sky cameras, among others and results in a better alternative to forecast variables in a minute scale [15] and Numeric weather prediction models (NWP) [8]. These predictions are suitable for the operation and control of power plants. For some applications, solar radiation predictions of 0 to 180 h are delivered online, every 6 h [11].

In addition, another advantage of knowing the future solar radiation lies in optimizing the control of solar energy in the electricity grid, which can ensure a favorable performance in the electricity generation market that may be used in the future in the Smart Grid field [16].

Currently, there is a lot of information published in journals about solar radiation prediction with ML. However, several ML methods about classification models to forecast variables are considered in the literature, such as artificial neural networks (ANN) or support vector machine (SVM) models and are commonly compared with other models. Additionally, forecasting solar radiation is a very complex phenomenon which can be influenced by many different factors and forecasting models tend to be more accurate. In this context, optimization algorithms are integrated to forecasting models to reduce error and improve its accuracy, so, it is necessary to present a review of prediction techniques with the use of search optimization algorithms (SOA) to improve prediction on different horizons. The objective of this article is to perform an analysis of solar radiation prediction techniques based on hybrid SVM and SOA models. The performance of SVM models is compared with conventional supervised learning models, such as artificial neural networks. SVM-SOA hybrid models are also analyzed to evaluate the accuracy of the predicted solar radiation data seen in the literature. It also presents the most suitable Kernel functions for hybrid models and finally, a general process flowchart of hybrid prediction techniques, according to the literature, is shown.

## 2. Solar Radiation Components

Modelling solar radiation is a very complex task because it is influenced by climatic zone, geographical area or seasons. Solar radiation provides the quantity of solar energy that reaches the Earth's surface during a particular time period [17]. It is important to know the whole phenomenon, starting from the definition of the sun, which is a star inside which a series of reactions take place that produce a loss of mass that is transformed into energy. Solar irradiance is the measured and recorded amount of energy that comes directly from the sun to the land surface [18]. There are three main types of solar radiation, which are the following: diffuse, direct and global solar radiation. Diffuse radiation is that which occurs

when radiation that comes directly from the sun is intercepted by the Earth's atmosphere, causing a scattering in the middle [19]. Diffuse solar radiation creates a problem in the generation of electricity by photovoltaic solar panels, reducing their generation capacity, that is, in solar radiation, clouds absorb all the incident energy and emit it again [20]. Direct solar radiation comes in a defined direction from the sun towards the earth, which, it is possible to concentrate it in a point for its use, however, it can also be reflected. This radiation is essential in the use and sizing of solar concentration systems [21]. With diffuse and direct solar radiation, it is possible to determine the global solar radiation in an area. Furthermore, it is possible to acquire information on global solar radiation by means of pyranometers that measure the solar irradiance from the sun to an area [22].

## 3. Solar Radiation Prediction Time Horizons

A prediction model essentially consists of a system of linear or non-linear equations that relate the future values of the variable to be predicted with the present and past values of the variable itself and the explanatory variables. Before making a prediction, you must define the prediction horizon on which the model should be applied [14]. The ML applications to predict solar radiations are becoming a trend in the transition of energy generation systems. The models are developed as a time series prediction problem, that will be solved as a classification model. Figure 1 visualizes the most used time horizons in prediction models using ML [23].

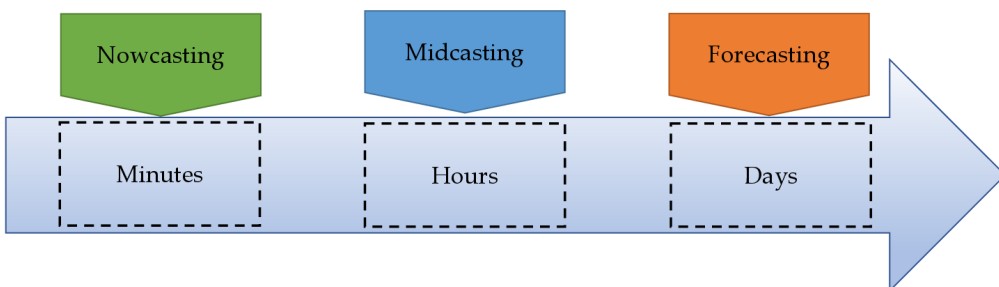

**Figure 1.** Prediction scale according to time horizon methods.

### 3.1. Time Horizon

The estimation of the output power of solar systems is necessary for the proper functioning of the electricity grid or for the optimal management of the energy flows that occur in the solar system. Before predicting the output of solar systems, it is essential to focus prediction on solar radiation. The prediction of components of solar radiation (global or diffuse) could be performed by several methods and the accuracy of a prediction model depends mainly on the time horizons [11].

For the development of any solar prediction model, we must contemplate the timescale (counted from a certain moment in which we make the prediction), which determines the future moment for which we make the predictions. The biggest drawback is its linear character, which makes it difficult for all problems to be properly modeled [22,24]. Models are very sensitive to unusual observations, forcing you to review the time series for detection and correcting before designing the prediction model [25].

### 3.1.1. Nowcasting Solar Radiation

The prediction horizon ranges from 15 min to a few hours, with no unanimity in its value [26]. In the short term, intra-hour forecasts are particularly useful for carrying out operations in the solar plant, balancing the grid, achieving automatic generation and trade control. Currently, it is very difficult to accurately predict solar radiation in the short term, as it involves knowing in advance how much energy solar plants produce instantly and this would be of great help in avoiding problems of supplying the line or avoiding surplus energy [27]. On this horizon, the statistical models that show the best performance are

those that use satellite images with greater accuracy in nowcasting predictions [15,28] and statistical models for solar radiation time series, such as NWP [29–31]. Likewise, in recent years, several studies have been presented on the use of Machine Learning (ML) for solar radiation prediction using vector support machines (SVM) that show better performance in classification and regression analysis in time series [32,33].

### 3.1.2. Forecasting Solar Radiation

Long-term predictions: correspond to a horizon above 48 or 72 h, reaching a limit of 7 days. The larger the horizon, the greater the prediction errors, making it difficult to make reliable predictions of atmospheric variables above those 7 days [34–36]. Time horizons represent prediction analysis, as, as has been observed in the literature, ML models are able to demonstrate that they are the best alternative in time series data analysis. However, the need to reduce error correction has been modified to make way for ML regression model models using search optimization algorithms (SOA) that require accurate selection of their parameters to improve their performance [37,38].

### 4. Support Vector Machine Models

Machine Learning (ML) is the process of learning to convert experience into expertise. A concise explanation is that a computer program learns from the experience from data recorded in a period of time; the performance of the program is evaluated in time by improving experience [39]. In other words, ML models interpret patterns through learning based on data obtained in a period through a training process to make a prediction, generating new data that will measure the behavior of a phenomenon in the future [40–42].

According to the literature, there are two main specific techniques that ML uses to develop learning methods using information directly from data acquired [43]. There exist two main learning methods: supervised learning that uses from present to previous data to forecast events [44,45] and unsupervised learning that analyzes the patterns from non-classified data and deduces a function to describe the behavior of the system [46]. Additionally, there are other learning methods that derive from these principal methods: semi-supervised learning and multiple instance learning [47]. In this article, the supervised learning using a support vector machine (SVM) will be discussed. The Figure 2 summarizes the two principal methods and its most used models to predict solar radiation.

According to [8], supervised algorithms are currently the most used, and within this method are neural networks and vector support machines. These models have been widely used in recent years, with vector support machines being recently integrated to provide new solar prediction techniques. In this context, SVM belongs to a technique called supervised learning defined as a technique for identifying the behavior between an input and an output variable [48].

The theoretical basis of SVM is to minimize the structural risk related to the empirical risk from the training process and the confidence range from Vapnik Chervonenkis dimension (VC dimension) [49]. The complexity of the problem will be reduced when the VC dimension is smaller, making the risk smaller [50].

The architecture of an SVM is a versatile and configurable model based on a kernel machine that could be treated as a classification or regression problem according to Vapnik equations [51], while a support vector regression is only used for regression problems [11]. Therefore, an SVM is a great alternative to solve classification problems in forecasting time series [52–55].

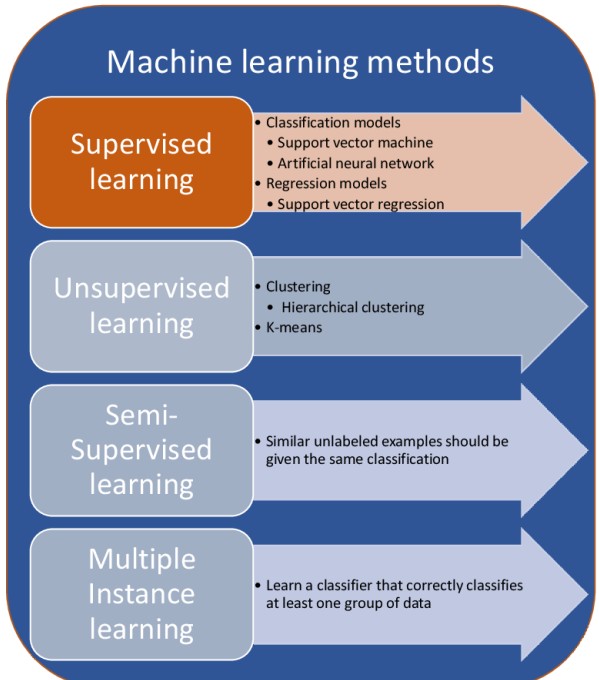

**Figure 2.** Machine learning techniques.

In order to predict a non-linear variable, such as solar radiation, it is necessary to have a data set $x_i$ that represents the input data sample space, $d_i$ which is the target value and to know the number of data points $n$ [14]; therefore, it is possible to resort to Vapnik's equations theory. The input variables to SMV models are related to the objective variable (variable to predict), visualizing the mapped data in a non-linear function $f(x)$ [51]:

$$f(x) = \omega \cdot \phi(x) + b \tag{1}$$

where $\omega$ is the normal vector, $b$ is a constant, also called bias term [56] and $\phi(x)$ is a large-dimensional spatial characteristic mapped by the space vector $x$. The coefficients $\omega$ and $b$ are calculated by minimization using the following optimization problem [24]:

$$R_{svm}(f) = C\frac{1}{N}\sum_{i=1}^{N} x_{i=1} = L_e(f(x_i), y_i) + \frac{1}{2}\|w\|^2 \tag{2}$$

$$L_e(f(x_i), y_i) = \begin{cases} \text{si } |f(x), y| - \epsilon & \text{for } |f(x), y| \geq \epsilon \\ 0 & \textit{otherwise} \end{cases} \tag{3}$$

where $\epsilon$ is a parameter of the model. $L_e(f(x_i), d_i)$ is the term that describes the $\epsilon$-th missing function, which indicates that errors below epsilon are not penalized, $d_i$ represents the solar radiation in the period $i$ and $C\frac{1}{N}\sum_{i=1}^{N} L_\epsilon(f(x_i), d_i)$ defines the empirical error of the SVM model. $\frac{1}{2}\|w\|^2$ is the regularization term, $C$ is the term that evaluates the error penalty function to regulate the compensation between the error or empirical risk and the term of regularization. The slack variables $\varsigma$ and $\varsigma^*$ indicate the excessive top and bottom skew, respectively. With these properties of the function to be optimized, it is possible to define Equation (2) as shown below [57]:

$$\text{minimize} \quad \frac{1}{2}\|w\|^2 + C\frac{1}{N}\sum_{i=1}^{N}(\varsigma_i + \varsigma_i^*) \tag{4}$$

$$\text{subject to: } = \begin{cases} |y_i - (\langle w, x_i \rangle + b) \geq \epsilon + \varsigma_i| \\ \langle w, x_i \rangle + b - y_i \leq \epsilon + \varsigma_i \\ \varsigma_i, \varsigma_i^* \geq 0 \end{cases} \tag{5}$$

To solve Equation (1), it is possible to use Lagrange and optimal constraints to obtain a non-linear regression function:

$$f(x) = \sum_{i=1}^{l} (\alpha_i - \alpha_i^*) K(x_i - x) + b \tag{6}$$

where $\alpha_i, \alpha_i^*$ are Lagrange multipliers. The term $k(x_i - x)$ is defined as the kernel function [55]:

$$K(x_i - x) = \sum_{i=1}^{D} \phi_i(x) + \phi_i(y) \tag{7}$$

The general architecture form of a SVM is shown in Figure 3.

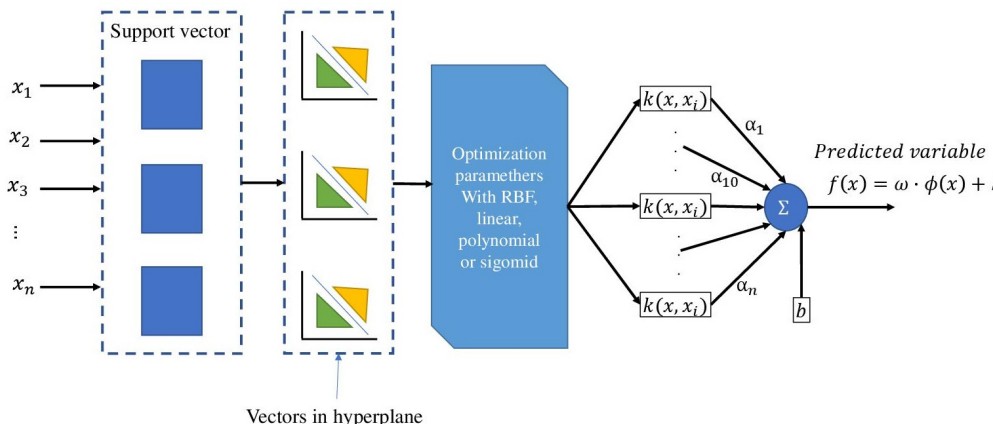

**Figure 3.** General architecture of a support vector maching (SVM) model according to [55].

### 4.1. Kernel Function Provided for SVM

SVM maps the data in a non-linear map to describe the linear process in the space of the predicted data according to the availability of data. This expression results in a simple linear combination problem attributed to the mapped space [58]. The kernel function allows a classification to be performed to form nonlinear boundaries to model complicated separating hyperplanes [59]. Figure 4 describes a projection from low to high dimension in space data.

The kernel parameters must be ideal to solve the problem classification according to the data to become separable in the next space. The four principal basic kernel functions are linear, polynomial, radial basis function (RBF) and sigmoid [57,60].

- Radial basis function (RBF): this function could perform nonlinear mapping of the samples into a higher dimensional feature space expressed by [61]:

$$K(x_i, x_j) = exp \left( -\frac{\|x_i - x_j\|^2 \sigma}{2} \right) \tag{8}$$

where $\sigma$ is the kernel weight and $x_i$ and $x_j$ are the inputs to the $i$-th and $j$-th dimensions, respectively.

- Linear kernel function: According to [57], the linear function to obtain the SVM parameters is described by the following:

$$K(x_i, x_j) = x_i \cdot x_j \tag{9}$$

- Polynomial kernel function: this is a typical example of a global kernel, defined as follows:

$$K(x_i, x_j) = (x_i \cdot x_j)^q \tag{10}$$

where $q$ is the degree of the polynom that will be used [62].

- Sigmoid kernel function: this function gives an explanation to practical viability [63]. This function is expressed as follows:

$$K(x_i, x_j) = tanh(v(x \cdot x_i) + c) \tag{11}$$

where $v$ and $c$ are adjustable kernel functions based on the data.

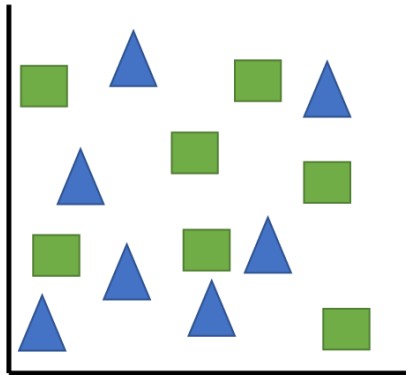
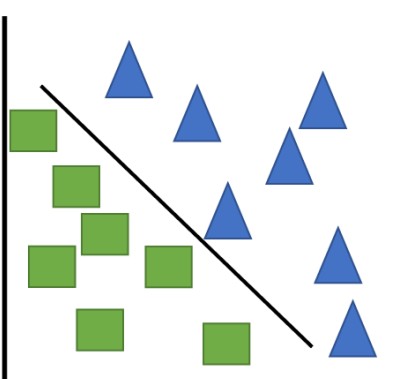

Input space                    Feature space

**Figure 4.** Mapping of the kernel function.

### 4.2. Search Optimization Algorithms

In general, SVM models and ANN models are widely used in forecasting power consumption and solar radiation because of its performance and easy adaptation for nonlinear variables. However, it is important to improve new methods to find the best fitting parameters of the model [64].

The importance of using genetic algorithms as a solution to the Vapnik model function (Equation (1)) is to find the most optimal point to be able to obtain better prediction results. Intelligent search algorithms (SOA) are techniques for searching and exchanging information between the individuals of a population, which will be the ones that can solve a non-linear optimization problem [65]. SOA models are frequently used in forecasting methods where it is quite crucial to determine weights coefficients to reduce the error [66].

### 4.3. Performance Evaluation

It is necessary to know the performance of the SVM model. To evaluate the performance of forecasting solar radiation data, statistical indicators were used [67]. According to the literature, there are five popular indicators that could determine the accuracy of the predicted data [68,69]:

- Mean absolute percentage error (MAPE): It is used to express the absolute error of the predicted and observed variables in percentage [70]:

$$MAPE = \frac{1}{m} \sum_{i=1}^{m} \left| \frac{y_i - \hat{y}_i}{y_i} \right| \tag{12}$$

- Root mean square error (*RMSE*) sizes the goodness of the fit related to forecast with high errors [71]:

$$RMSE = \sqrt{\frac{1}{m}\sum_{i=1}^{m}(y_i - \hat{y}_i)^2} \qquad (13)$$

- Mean bias error (*MBE*) indicates the deviation of predicted data from the observed data to provide the performance information of short and long-term use of the model [72]:

$$MBE = \frac{1}{m}\sum_{i=1}^{m}(y_i - \hat{y}_i) \qquad (14)$$

- Mean absolute error (*MAE*) indicator gives a perspective of the performance of the prediction model by viewing how close the predicted variables are to observed variables [73]:

$$MAE = \frac{1}{m}\sum_{i=1}^{m}|y_i - \hat{y}_i| \qquad (15)$$

- Relative root mean square error (*RRMSE*) quantifies the relative spread in the error [74,75]

$$RRMSE = \frac{1}{\bar{y}}\sqrt{\sum_{i=1}^{m}\frac{(y_i - \hat{y}_i)^2}{N}} \qquad (16)$$

where $y_i$ is the global solar radiation measured, $\hat{y}_i$ is the predicted global solar radiation, $\bar{y}$ is the mean global solar radiation, $m$ is the number of forecast data points and $N$ is the number of validation data. These indicators make it possible to know the efficiency of the SVM models: If the statistical indicator value is zero in the ideal case and presents a good performance if is closer to zero [8,57,67,76].

## 5. Results in Hybrid Techniques: Support Vector Machine with Search Algorithms Review

This work compiled and analyzed scientific articles focused on the construction of solar radiation prediction models with a hybrid method based on support vector machine and search optimization algoritms. The articles that are studied show the advances in this field and the solutions proposed to obtain predictive radiation data with the minimum error calculated with the statistical indicators.

SVMs relate to regularization networks and offers an advance on the ANN model. It is based on the theory of statistical learning that adopts least squares methods to solve the problem to least square solutions through a set of linear equations based on the minimization of structural risk.

Therefore, the SVM model can avoid excessive adjustment of the training data, does not require an iterative adjustment of the model parameters, has better generalization, requires few cores and has good performance [77]. To determine the appropriate choice of the prediction model with a support vector machine, the climatic variables of maximum and minimum air temperature ($T_{max}$ and $T_{min}$, respectively), maximum and minimum relative humidity ($H_{max}$ and $H_{min}$, respectively), wind speed ($w_s$), evaporation ($E$) and vapor pressure estimates ($V$) were the most used for future global solar radiation $G_h$ [78–81]. This model can avoid the excessive adjustment of the training data, does not require an iterative adjustment of the model parameters, has better generalization, requires few cores and has good performance [55]. Figure 5 visualizes the general process to predict global solar radiation using the climatic variables.

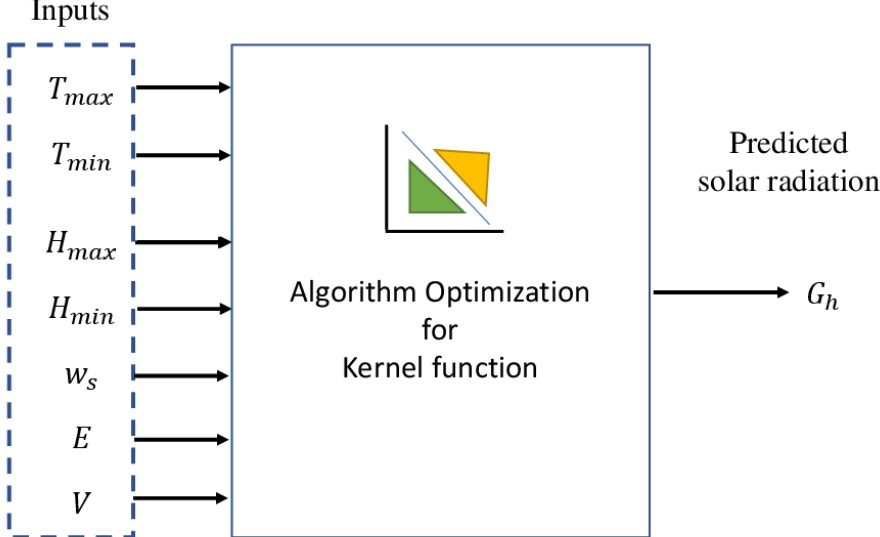

**Figure 5.** Training process using climatic variables for SVM models.

Different researchers have performed models based on SVM with no optimization models, and the results were prominent. Ref. [76] presented a hybrid SVM model in predicting solar radiation on two sites for PV panel surfaces using diffuse, direct and global solar radiation as input variables. The results show that the SVM model improves the prediction compared to the ANNs in the training phase with a maximum time of 0.0468 s, almost 5 times faster than an RNA model and 2 times faster in the testing phase, with 2.15 s. The stability of the model in this application lies in the use of these three radiation components with an RMSE three times smaller than an ANN, varying in a range from 18.34 to 31.15.

Authors from [36] proposed a solar prediction model based on SVM for one-hour ahead based on the typical climatic variables. They mentioned that SVM regression significantly improves the prediction accuracy according to the statistical indicators.

Authors from [67] proposed multiple SVM models using temperature equations with different kernel equations using only temperature variables from meteorological stations. The researchers observed that creating several empirical temperature-based equations and evaluating the models by several statistical indices, the prediction improves significantly in its performance.

In [82], a forecasting model based on SVM and using satellite images of clouds as the input space was presented. They used 4-year registers of cloud monitoring systems configured to perform the model by using large-scale data in multiple inputs and outputs. The performance SVM model was compared to other predicting models, providing a robust and great alternative to predict solar energy with $RMSE$ 10.86.

In [83], a model was developed to estimate global solar radiation based on a support vector machine (SVM) using an index that indicates air quality as an input variable to evaluate the performance of this technique. Compared to existing models, such as neural networks, the model presents a great performance in the accuracy of global solar radiation models by using an air quality indicator as an additional input parameter. The related works only used SVM models. Table 1 summarizes the findings about using SVM as an important alternative to ANN or other models to predict solar radiation.

**Table 1.** Techniques of solar radiation prediction using SVM.

| Reference | Analysis of Results | Time Horizon | Kernel Function | MAPE | RMSE | MBE | MAE | RRMSE |
|---|---|---|---|---|---|---|---|---|
| [76] | Compared ANN and SVM in predicting the solar radiation | 1 day | polynomial | - | 28.39 $\frac{W}{m^2}$ | - | - | - |
| [83] | SVM performs better than other models if air quality index is used in the models using polynomial kernel function | 1 h | Polynomial | 8.24% | - | - | - | - |
| [36] | They compared the SVM with (ANN) and Non linear autoregressive (NAR), showing that SVM performs well in prediction accuracy | 1 h | RBF | - | 4.26 $\frac{W}{m^2}$ | - | - | - |
| [82] | SVM performs great than ANN models and can be effectively used for grid operations and energy management systems | 15 min | RBF | - | 28.00 $\frac{W}{m^2}$ | - | - | - |
| [67] | Polynomial kernel function performs great using temperature equations in SVM models | 1 h | Polynomial | - | 0.83 $\frac{MJ}{m^2}$ | - | - | 9.00% |

In recent years, the state-of-the-art with respect to SVM has been demanding due to its high performance, so various methods of improving these models have been developed, such as integrating a search algorithm to the SVM model. Some works have proposed various hybrid techniques based on the flow diagram in the Figure 6.

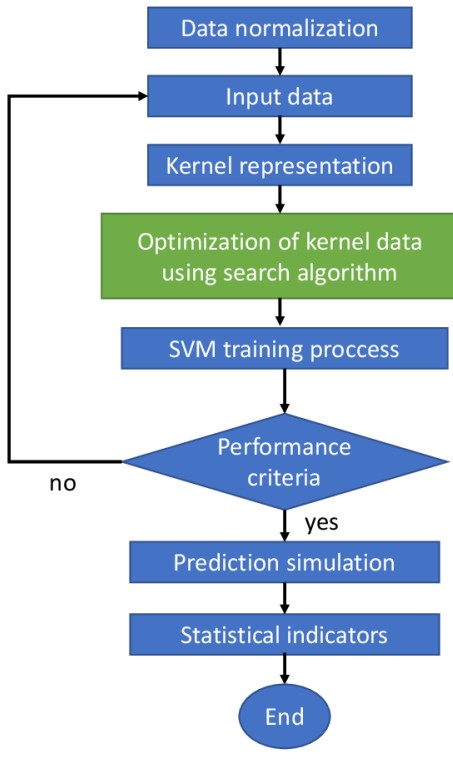

**Figure 6.** Flow chart of SVM model with search algorithm.

The work of [14] showed an SVM with the Firefly Algorithm (FFA) to estimate future solar irradiance. A Firefly Algorithm (FFA) was used to calculate the best kernel parameters of the SVM model to obtain the most accurate prediction data. The model presents a better performance compared to ANN, which has RMSE = 1.86% and a MAPE value of 11.51%. The results are promising for the use of FFA with SVM as a great alternative to predicting the global component of the solar radiation. Additionally, they specified a risk in zones with no constant climatic conditions.

In [84], a new approach was developed using data mining techniques to model and improve the prediction of hourly global solar radiation the next day. They report an RMSE value of 23.5% when the independent variables were the values of the meteorological parameters of the previous day and an RMSE of 22.9% when the independent variables were the forecasts of meteorological variables for the same day to be forecast.

The authors [85] developed an SVM model optimized by glowworm swarm optimization (GSO) to estimate solar radiation. Employing the Eclat algorithm (data mining) to choose the appropriate predictors, the support vector machine (SVM) with penalty function of the model structure and the GSO algorithm to improve forecast accuracy, selecting the appropriate parameters. The tests were carried out in four areas of the USA using the input variables of average daily airmass, dew point, relative humidity, opaque cloud cover, wind speed, vertical wind shear, pressure, albedo, zenith angle, azimuth angle, net radiation, global normal radiation and global extraterrestrial radiation. The results show a great performance—RMSE = 0.43 $\frac{W}{m^2}$ and MAPE 5.64%.

In [86], an SVM-FFA model for the prediction of the monthly mean daily horizontal global solar radiation in the port of Bandar Abbass was proposed, located in the southern coastal region of Iran. They also show that their model is highly efficient in estimating the monthly mean daily horizontal global solar radiation. With the following statistical indicators, MAPE = 3.3252.

Ref. [87] presented an SVM model with optimization using GA to compare its performance with a grid search evaluating the accuracy by the analysis of the parameters of the four kernel functions. They found that the grid search could be a good alternative only if a low dimensional dataset, which is only present in few cases of classifications problems, but GA is the best alternative in more cases, presenting stability above 15.9-times that of a grid search.

Research from [57] presented a study of three models: adaptive neuro-fuzzy inference system (ANFIS), ANN and (SVM) models using GA in the three cases to estimate the kernel model parameters in order to compare their accuracy and performance in daily global solar radiation. The results show a huge advantage using SVM-GA to predict solar radiation. They found that SVM-GA also determined the best kernel parameters by evaluating the error to the global minimum convergence.

In [55], a wavelet-coupled vector support machine model for predicting global solar incidents using a limited meteorological data set for the city of Brisbane, Australia was performed. The data were decomposed into a subset of wavelets, transforming the input space into discrete variables to create new time series using the Daubechies-2 wavelet to a detailed level. This hybrid model obtained an approximation of $R = 0.965$. The input variables used in this model are the basic climatic variables and add hours of sunshine *St* to predict the daily global incident solar radiation ($R_n$). They also mentioned that climatic anomalies could make the W-SVM model less accurate for solar radiation forecasting.

The authors from [88] presented a GA-SVM model to predict the short-term power forecasting of a PV system on a residential scale. GA was used to find the optimal parameters values for a kernel within a base classifier of the SVM. They used climatic variables and added the solar radiation to predict the power load of a photovoltaic system, that shows that SVM is multi-configurable according to the problem. The results show that GA-SVM outperforms the conventional SVM. The GA shows its great performance by finding a local minimum that is translated in a high-accuracy solar radiation forecasting. Table 2 summarizes the findings on the use of SVM optimized with search algorithm techniques.

Ref. [89] performed a comparison between SVM models and copula-based nonlinear quaintly regression (CNQR) to predict dayli diffuse solar radiation and mentioned the principal risk working with SVM models: There is a possibility that SVM could not perform an accurate prediction if lower parameters are considered in the space data; with more input parameters, the accuracy will increase but the parameter optimization increases considerable, leading to a high cost for the prediction.

The authors from [90], developed three hybrid models to predict the daily solar radiation in urban zones: SVM-PSO model, Bat algorithm with SVM and Whale optimization algorithm with SVM using the general climatic data and adding the Ozone ($O_3$) in the space variables. They found that the $O_3$ variable improves the accuracy of the predicted data compared to other parameters such as air pollution, using the RMSD statistical indicator to measure the performance, with values of 11.1%, 10.0% and 10.4%, respectively, for the three developed models.

**Table 2.** Techniques of solar radiation prediction using SVM and search algorithms.

| Reference | Analysis of Results | Time Horizon | Optimization Model | Kernel Function | MAPE | RMSE | MAE | RRMSE |
|---|---|---|---|---|---|---|---|---|
| [14] | SVM-FFA present a better performance in comparisson with ANN models | 1 h | FFA | RBF | 11.51% | - | - | 1.86% |
| [84] | Mining data to forecasting hourly global solar radiation | 1 h | SVM-R | Sigmoid | - | 119 $\frac{W}{m^2}$ | 79 $\frac{W}{m^2}$ | 22.90% |
| [85] | Evaluated the performances of SVM, HARD-RIDGE-SVM, SVM-HARD and GSO-SVM-HARD model | 1 day | GSO | Hilbert space | 5.64% | 0.43 $\frac{W}{m^2}$ | - | - |
| [86] | SVM-FFA present a better performance compared to the ANN, GP, and ARMA techniques | 1 month | FFA | RBF | 3.32% | 0.18 $\frac{kW}{m^2}$ | - | 3.73% |
| [87] | SVM parameter optimization using GA is more than 15.9 times faster than using grid search. F-measure was used to evaluate the performance | - | GA | RBF sigmoid | 8.24% | - | - | - |
| [55] | SVM incorporates a discrete wavelet transformation algorithm for pre-processing of inputs | 1 day | Wavelet | RBF | 4.69% | 1.18 $\frac{MJ}{m^2}$ | 0.92 $\frac{MJ}{m^2}$ | 5.94% |
| [57] | SVM-GA models has higher prediction accuracy in tropical warm sub-humidthan ANN model | 10 min | GA | - | - | 2.57 $\frac{MJ}{m^2}$ | 1.97 $\frac{MJ}{m^2}$ | - |
| [88] | GA-SVM outperform SVM classical models in classification climatic data | 1 h | GA | Gaussian | 1.70% | 11.22 $\frac{W}{m^2}$ | - | - |
| [89] | SVM-FFA better than copula-base nonlinear quantile regression | 1 day | - | RBF | - | 1.18 $\frac{MJ}{m^2}$ | - | 18.00% |
| [90] | Mentioned that SVM with PSO convergence to local optimal solution faster than the other proposal algorithms. | 1 h | PSO | RBF | - | - | 0.99 $\frac{MJ}{m^2}$ | 2.90% |

## 6. Discussion and Conclusions

This article presented a state-of-the-art in solar radiation prediction techniques through the use of support vector machine models (SVM) and the optimization of the search for the parameters that guarantee a favorable performance in the accuracy of the forecasting data.

It was observed that SVMs by themselves show a better performance in predicting solar radiation than artificial neural networks and other prediction models such as auto regression. In this context, it is possible to define these models as premature techniques, which today are consolidated as the best predictive models, since these models emerged 6 years ago [8]. The SVM models show an improvement when evaluating the polynomial kernel functions using only as a basis the climatic variables of maximum and minimum air temperature ($T_{max}$ and $T_{min}$, respectively), maximum and minimum relative humidity ($H_{max}$ and $H_{min}$, respectively), wind speed ($w_s$), evaporation ($E$) and vapor pressure estimates ($V$). That base of the input space is sufficient to perform a prediction model based on SVM. However, it is possible that the polynomial function demands a high cost in performance because it can increase in degree, so another alternative is the use of a radial basis function (RBF). According to the literature, RBF to estimate the kernel parameters can improve the accuracy up to 7% compared to higher degree polynomial functions. Likewise, the most widely used validation method is $RMSE$, since the observed error contributes to the tuning of the weights of the SVM model. Additionally, according to [76], SVM could be 5 times faster than ANN in the training phase and 2 times faster in the testing phase.

The execution time of each optimization algorithm will depend on the number of input variables to the prediction model, the number of data and the prediction horizon. In order to obtain a more robust model, it must be calibrated with the variables and geographical conditions, since they change according to the area, so the accuracy totally depends on the location. These search algorithms considerably revolutionize the performance of the SVM model compared to a simple SVM model, contributing to obtaining a lower prediction error in the output of the solar radiation data. In addition, in these models, the RBF kernel function presents better performance in the search for parameters due to the inclusion of these search algorithms. These algorithms can be evaluated based on MAPE, RMSE and MAE as the main statistical indicators to evaluate the predicted data of solar radiation that measure the performance of the model.

Finally, the versatility of the model using only climate data for the prediction of solar systems is noteworthy, such that in the future they may be a beneficial alternative in the sizing, generation and management of alternative energy to contribute to the energy transition that in the coming years will be one of the main issues that researchers will tackle to solve problems.

**Author Contributions:** Conceptualization, J.M.Á.-A. and J.G.R.-M.; investigation, J.M.Á.-A.; validation, G.R.-L., E.V.-R.J. and S.A.O.-B.; writing—original draft preparation, J.M.Á.-A.; discussion, J.G.R.-M.; writing—review and editing, M.T.-P. and J.G.R.-M.; formal analysis, J.G.R.-M. All authors have read and agreed to the published version of the manuscript.

**Funding:** This research received no external funding.

**Institutional Review Board Statement:** Not applicable.

**Informed Consent Statement:** Not applicable.

**Data Availability Statement:** The data presented in this study are available on request from the corresponding author.

**Conflicts of Interest:** The authors declare no conflict of interest.

## Abbreviations

The following abbreviations are used in this manuscript:

| | |
|---|---|
| $NWP$ | Numerical weather prediction |
| $ML$ | Machine learning |
| $SVM$ | Support vector machine |
| $SOA$ | Search optimization algorithms |
| $G_h$ | Global solar radiation |
| $f(x)$ | Vapnik Function |
| $x$ | Input data |
| $\omega$ | Normal vector |
| $b$ | Bias term |
| $\phi(x)$ | Large-dimensional spatial characteristic |
| $(C, \epsilon)$ | parameters of the model |
| $\varsigma$ | Excessive top skew |
| $\varsigma^*$ | Excessive buttom skew |
| $K$ | Kernel |
| $\alpha$ | Lagrange multiplicators |
| $i$ | index |
| $\frac{1}{2}\|w\|^2$ | Regularization term |
| $RBF$ | Radial basis function |
| $q$ | Degree of polynom |
| $GA$ | Genetic Algorithm |
| $PSO$ | Particle Swarn Optimization |
| $MAPE$ | Coefficient of determination |
| $RMSE$ | Root mean square error |
| $MBE$ | Mean bias error |
| $MAE$ | Mean absolute error |
| $RRMSE$ | Relative root mean square error |
| $T_{max}$ | Maximum temperature |
| $T_{min}$ | Minimum temperature |
| $H_{max}$ | Maximum relative humidity |
| $H_{min}$ | Minimum relative humidity |
| $w_{max}$ | Wind speed |
| $E_{min}$ | Evaporation |
| $V$ | Vapor pressure |

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
