# Peer review of "Hybrid Techniques to Predict Solar Radiation Using Support Vector Machine and Search Optimization Algorithms: A Review"

_applsci, doi:10.3390/app11031044_

Round 1

Reviewer 1 Report

Dear Authors,

Thank you for your effort in conducting the review and the manuscript. The objective is clear and the paper is structured accordingly. The paper is easy to read.

However following are a few comments for you to consider in order to improve the manuscript:

  1. The language in the paper needs significant improvements. For example, line 13 "...SVM with GA overperformed" the appropriate word is outperformed. The meaning of a sentence thereby can be misinterpret if appropriate terminology is not used.
  2.  Line 28, what does it mean "not constant"? PV plant of 1 MW capacity is constant, is it not?
  3. Figure 2 is limited to a few algorithms. The authors might acknowledge that there are many other types of algorithms including that of a mix between supervised and unsupervised. Thereby this reviewer would advise to consider writing that this paper is focusing on only certain types of algorithm while the list expands beyond what is presented. 
  4. RMSE is normally known as Root Mean Square Error not relative RMSE. Equation 15 and 13 have the same abbreviation. Please modify. 
  5. There are only 6 papers compared in table 2. This reviewer was expecting an extensive comparison among at least 20 evolutionary algorithm implementations. 
  6. Please also consider the convergence time or number of iterations metric which is usually associated with a heuristic technique. 

The review is limited by a very specific selection of papers which is fine but a journal review paper should include extensive comparison among the methods. For example proposing a criteria matrix for comparison could be interesting for the readers to better understand which method to choose for a specific prediction task.  

With thanks,

Reviewer

Author Response

Reviewer#1: Thank you for your effort in conducting the review and the manuscript. The objective is clear and the paper is structured accordingly. The paper is easy to read. However, following are a few comments for you to consider in order to improve the manuscript:

Author response:  We appreciate your review of our paper. All of your concerns were attended.

Author action: We updated the manuscript by following all your observations.

Reviewer 2 Report

Dear Authors,

The manuscript concerns the state-of-the-art of solar radiation prediction be means of SVM. It is clearly within the scope of the Journal and I encourage the Authors to continue to develop their work for future publication. Unfortunately I cannot recommend publication of the article in its current form. The reasons for this recommendation are (they are not listed in order of priority):

- The article seems chaotic. There are several repetitions of the same thought within one section (lines 173-184). The figures are prepared in an inconsistent manner. For example, the arrows in the diagrams (Figures 2, 5, 6, 7, 8) are drawn in different styles.

- Division of the introduction section into 1 sub-section is unnecessary.

- The description of the prediction horizon is unclear (lines 51-53).

- Does the division of models into classes of nowcasting, midcasting and forecasting refer only to prediction of solar radiation, weather or perhaps prediction models at all (Figure 1)?

- Please provide the explanation of abbreviation before they appear in the text (line 103).

- You use the same symbols for different variables, e.g. E as experience (line 111) and evaporation (line 226).

- There is no need to introduce symbolic names (experience E, performance of program P) if you never use them later.

- If Figure 3 shows the general architecture of the SVM model, the model output should not be marked as solar radiation.

- I see no reason to describe the mechanisms of GA and PSO. These are generally known methods of optimisation. In addition, in this case they are only used as a tool to select the parameters of the SVM model. Particularly so as you refer in a later part of the article to work using not only GA and PSO but also FFA and Wavelet (Table 2).

- The explanation of statistical indicator is not consistent. You mentioned maximum value of indicators is 1, while Tables 1 and 2 show a couple examples which they exceed this limit.

- I noticed a difference between RMSE value mention in the text (line 248) and in the Table 1.

- I think that the number of works analysed is insufficient for the standards of the review article. I admit that the bibliography is very rich, but the results section only contains a reference to 5 works in one variant and 6 in another. What is the outcome of this comparison and how can it be made? For example, in the paper [76] MAPE and RMSE were used to assess the model, while in [75] IEA and RRMSE. Is it possible to answer the question which Kernel function (linear or RBF) is better or which of the optimisation methods (GA or PSO) is more effective? I think not. I also lacked an analysis of what input signals the authors of particular works used in their models. Perhaps it is worth presenting simple statistics which signals are most popular. Another extremely important question is why this set of signals was chosen. Solar radiation is a very complex phenomenon which can be influenced by many different factors, such as geographical area, season, climate zone, etc. Answers to such questions should be the subject of meta-analysis in the review article.

I hope you find my comments useful and they help you to make further improvement of your paper.

Best wishes,

Reviewer

Author Response

Reviewer#2: The manuscript concerns the state-of-the-art of solar radiation prediction be means of SVM. It is clearly within the scope of the Journal and I encourage the Authors to continue to develop their work for future publication. Unfortunately I cannot recommend publication of the article in its current form. The reasons for this recommendation are (they are not listed in order of priority):

Author response:  Thank you for your comments.

Author action: We updated the manuscript by following all your observations.

Reviewer 3 Report

The authors attempt to review the hybrid support vector machine (SVM) technique applied to the forecast of global solar irradiance. The manuscript requires a major revision before a new review. Below are the main points that the authors must take into account when writing the revised version.

  1. The main problem with this manuscript is the language. The manuscript is hard to read and in some places even incomprehensible. English must be carefully checked before a resubmission. My advice for the authors is to use several short sentences instead of long sentences wherever possible.

The first sentence in Abstract is an example: “The use of intelligent algorithms for global solar prediction is an ideal tool for research focused on the use of solar energy.” What does “global solar prediction” mean?

  1. There are many errors that should be corrected. For example:

- the same first phrase in Abstract: The authors probably refer to forecasting solar irradiance or solar irradiation.

- second phrase in abstract: Is solar radiation really the forecasted physical quantity? Or is it solar irradiance/solar energy? What is the difference between solar radiation and solar irradiance / irradiation / energy?

- line 37-40 systems.  It seems that there is a confusion between estimation and forecasting.

- line 41: „Solar prediction plays an important role in research today” !?

- line 59: „Prediction models are attractive to governmental and non-governmental agencies.” !?

- line 61: The definition of the forecasting model is confused. See the definition in the cited papers.

Line 74: „For the development of any solar prediction model we must contemplate, from the first moment  of design, the horizon of prediction, that is, the period of time (counted from a certain moment in which we make the prediction), which determines the future moment for which we make the predictions.” !?! What is meaning of this phrase? Contemplate what?

and so on. The authors must correct the manuscript by making it less vague and more technical.

  1. A section devoted to solar radiation modeling is required. In this section notions such as solar irradiance, solar irradiation, global, diffuse and beam components must be defined.
  2. Line 92: What does the title “Forecast prediction” mean?
  3. Line: Is MAPE really the coefficient of determination?
  4. Line 230: Are SVM models running with climatic variable at the input?
  5. Table 2 seems to contain some errors. Is RMSE expressed as percent? What is the difference between RMSE and RRMSE? Generally, explanations for the values inserted in the tables are required.
  6. Line 100. The objective of this work is insufficiently motivated. It must be clearly presented. What are the novelties reported in this study compared to others published in the last years? What does this study bring in addition?

Author Response

Reviewer#3: The authors attempt to review the hybrid support vector machine (SVM) technique applied to the forecast of global solar irradiance. The manuscript requires a major revision before a new review. Below are the main points that the authors must take into account when writing the revised version.

Author response:  We appreciate your review of our paper. All of your concerns were attended.

Author action: We updated the manuscript by following all your observations.

Reviewer 4 Report

Review articles are always welcome in the scientific literature, as they provide a comprehensive and elaborate vision on a specific topic. They are an invaluable help for novice researchers and even for experienced ones, as they provide a detailed and current vision on a specific topic.

Author Response

Reviewer#4: Review articles are always welcome in the scientific literature, as they provide a comprehensive and elaborate vision on a specific topic. They are an invaluable help for novice researchers and even for experienced ones, as they provide a detailed and current vision on a specific topic.

Author response:  We appreciate your support in this manuscript. Thank you.

Round 2

Reviewer 2 Report

I am grateful to the authors for the changes they have made. In my opinion, the article has been significantly improved. The discussion section is still a bit too short but it already contains the information I mentioned in the previous review, e.g. the choice of regressors in the individual papers.
The article in its current form is ready for publication.